# Malignant Acute Colonic Obstruction: Multidisciplinary Approach for Endoscopic Management

**DOI:** 10.3390/cancers16040821

**Published:** 2024-02-18

**Authors:** Aurelio Mauro, Davide Scalvini, Sabrina Borgetto, Paola Fugazzola, Stefano Mazza, Ilaria Perretti, Anna Gallotti, Anna Pagani, Luca Ansaloni, Andrea Anderloni

**Affiliations:** 1Gastroenterology and Endoscopy Unit, Fondazione IRCCS Policlinico San Matteo, Viale Camillo Golgi 19, 27100 Pavia, Italy; a.mauro@smatteo.pv.it (A.M.); a.anderloni@smatteo.pv.it (A.A.); 2Department of Internal Medicine, PhD in Experimental Medicine Italy, University of Pavia, 27100 Pavia, Italy; 3Medical Oncology Unit, Fondazione IRCCS Policlinico San Matteo, 27100 Pavia, Italy; sabrina.borgetto01@universitadipavia.it (S.B.); a.pagani@smatteo.pv.it (A.P.); 4Department of General Surgery, Fondazione IRCCS Policlinico San Matteo, 27100 Pavia, Italy; p.fugazzola@smatteo.pv.it (P.F.); l.ansaloni@smatteo.pv.it (L.A.); 5Institute of Radiology, Fondazione IRCCS Policlinico San Matteo, 27100 Pavia, Italy; ilaria.perretti01@universitadipavia.it (I.P.); a.gallotti@smatteo.pv.it (A.G.)

**Keywords:** acute colonic obstruction, colorectal cancer, endoscopic stent, bridge to surgery, antiangiogenic agents, self-expandable metal stent, CT scan

## Abstract

**Simple Summary:**

Acute colonic obstruction is one of the most common manifestations of locally advanced colorectal cancer. Endoscopic stenting has become by far the minimally invasive treatment of choice for malignant colonic obstruction especially in the palliative setting. However, there are still controversies in the literature about the usefulness and safety of endoscopic stenting as a bridge-to-surgery approach or in patients on antiangiogenic therapy. Moreover, endoscopic colonic stenting is an operative procedure that requires adequate pre-interventional management and specific endoscopic knowledge. The present review aimed to summarize the optimization of endoscopic management of patients with malignant acute colonic obstruction based on a multimodal connection with the various medical specialties involved in managing this urgent clinical scenario.

**Abstract:**

Patients presenting with acute colonic obstruction are usually evaluated in the emergency department and multiple specialties are involved in the patients’ management. Pre-treatment evaluation is essential in order to establish the correct endoscopic indication for stent implantation. Contrast-enhanced imaging could allow the exclusion of benign causes of colonic obstruction and evaluation of the length of malignant stricture. Endoscopic stenting is the gold standard of treatment for palliative indications whereas there are still concerns about its use as a bridge to surgery. Different meta-analyses showed that stenting as a bridge to surgery improves short-term surgical outcomes but has no role in improving long-term outcomes. Multidisciplinary evaluation is also essential in patients that may be started on or are currently receiving antiangiogenic agents because endoscopic stenting may increase the risk of perforation. Evidence in the literature is weak and based on retrospective data. Here we report on how to correctly evaluate a patient with acute colonic malignant obstruction in collaboration with other essential specialists including a radiologist, surgeon and oncologist, and how to optimize the technique of endoscopic stenting.

## 1. Introduction

Colorectal cancer (CRC) is the third most frequently diagnosed malignancy in the world and the second leading cause of cancer-related mortality [1]. About 10–40% of colorectal cancer patients have bowel obstruction at the time of diagnosis, particularly on the left side [2], and large bowel obstruction (LBO) is a common condition that accounts for about 24% of admissions for acute mechanical bowel obstruction [3]. Acute colonic obstruction requires urgent management in order to avoid further complications such as perforation or ischemia. In the past, urgent surgery with colonic resection and/or stoma formation was the only available treatment. Endoscopic decompression with the application of self-expandable metal stents (SEMS) within the stricture has been proposed in the last decades as a less invasive option and has become the treatment of choice for patients who need palliative treatment [4].

The surgeon is usually the first specialist to evaluate patients with acute colonic obstruction in the emergency department; the consultation with the endoscopist aims to share the indication of endoscopic stenting, especially for resectable patients for whom there are still no clear indications for the bridge-to-surgery stenting [5]. When evaluating the patient for colonic stenting, the on-call endoscopist usually performs a virtual urgent multidisciplinary consultation with the main figures involved in the management of the patient. Radiological consultation is then essential before endoscopic stenting in order to confirm the malignant etiology of the acute obstruction, to better define the colonic anatomy and to identify urgent criteria for the timing of endoscopic decompression. Lastly, acute colonic obstruction may develop in patients with a known colon cancer under active chemotherapy treatment. Antiangiogenic therapy (e.g., bevacizumab) has been associated with an increased risk of perforation in patients treated with colonic stenting [6]. However, evidence is lacking and based on retrospective studies. Thus, multidisciplinary evaluation with an oncologist and surgeon is crucial in order to identify the correct management strategy and to discuss the risks and benefits of the endoscopic procedure.

Endoscopic colonic stenting is an interventional procedure that requires specific endoscopic skills and experience in the interpretation of intraprocedural radiological frames [7]. The placement of a colonic stent is usually easy in the case of short and linear strictures, whereas it could become particularly challenging in the case of difficult anatomic locations, including lesions close to the anal verge, in the right colon or colonic flexures.

The present review intends to assess the appropriate clinical indications for colonic stenting in a multidisciplinary context and to provide practical tips and tricks for the endoscopic procedure.

## 2. Surgeon–Endoscopist Collaboration for Indication of Endoscopic Stenting

Acute LBO remains a surgical urgency and initial evaluation should be performed by the general surgeon. Preliminary surgical evaluation is aimed at confirming the diagnosis of malignant etiology, excluding other benign causes of LBO that usually do not require endoscopic management except for specific cases (e.g., endoscopic decompression for sigmoid volvulus). Surgical consultation is also essential for the interpretation of CT scan images in order to exclude the presence of abdominal complications that require urgent surgical therapy such as bowel perforation and/or ischemia.

Once a diagnosis of malignant LBO is confirmed, it is essential for the surgeon and endoscopist to collaboratively determine the indication for endoscopic stenting. Endoscopic treatment is sometimes contraindicated when there are signs of colonic ischemia or perforation and therefore emergency surgery (ES) is the only possible treatment [8]. ES contemplates the emergency resection of the primary lesion with an immediate colorectal anastomosis (possibly associated with a diverting loop ileostomy) or without a prompt recanalization and the creation of a colostomy (“Hartmann’s procedure”). When contraindications to stenting are excluded, it is important to evaluate the clinical context of the patient. Patients unsuitable for surgery due to advanced disease (e.g., metastatic CRC) or for the presence of multiple comorbidities should be referred for endoscopic palliative stenting. When the patient has a resectable CRC, it is possible to perform a two-step approach consisting of the endoscopic placement of a SEMS to resolve the obstruction and in the elective surgical resection a few weeks after (stent as a bridge to surgery).

Acute colonic obstruction may also be caused by a non-primary colonic tumor such as pelvic tumors, advanced gastric or other metastatic cancers that cause extrinsic compression. Usually, patients with extrinsic obstruction have milder symptoms than patients with strictures caused by primary colonic cancer [5]. In this subgroup of patients, the aim of the treatment is usually palliative and endoscopic stenting, which demonstrates feasibility but with lower rate of clinical success [9]. Surgical treatment should be evaluated according to the performance status of the patients and to the resectability of the primary neoplasia.

### 2.1. Palliative Colonic Stenting

In patients with LBO due to colorectal cancer and advanced/metastatic disease that are not eligible for curative treatment, surgical or non-surgical palliation should be considered. Resection, bypass, and colostomy are the available surgical options, but the European Society of Gastrointestinal Endoscopy (ESGE) Guideline [10] strongly recommends colonic stenting as the preferred treatment for the palliation of malignant colonic obstruction. Some studies comparing colonic stenting and ES for palliation [11,12,13,14,15,16,17,18] showed a that technical success of stent placement ranged from 88% to 100%, while the initial clinical relief of obstruction was significantly higher after palliative surgery compared to colonic stenting [12]. Conflicting results have been reported regarding short-term mortality and overall morbidity [11,12,13,14,15], but colonic stenting was associated with a shorter length of stay, lower costs, a lower intensive care unit admission rate and a shorter time to the initiation of chemotherapy [11,12,13,14,15,17,18]. Furthermore, patients treated with endoscopic stenting had better quality of life if compared to patients treated with palliative surgery until 12 months after the procedure [18].

However, some observational studies [9,19,20] showed lower technical success and an increased complication rate for colonic stenting in patients with peritoneal metastases, because the main limitations to the success of bowel stenting are the presence of multiple sites of obstruction. In this situation, a surgical approach could be considered.

### 2.2. SEMS Role as Bridge to Surgery

Placement of an SEMS before elective surgery has the rationale to allow resolution of the obstruction and consequently to obtain patients’ stabilization, improvement of general conditions and nutritional status, accurate staging and definition of a tailored treatment for the patients [21]. As a result, higher quality oncologic resections could be performed using minimally invasive approaches and without the need for permanent stoma [22]. However, some controversies have emerged regarding the oncological safety of SEMS. It has been speculated that the increased interstitial pressure in the neoplastic mass can cause cell dissemination, cell shedding and tumor embolization into lymphatic vessels, as a consequence of a higher rate of recurrence observed in patients with SEMS [23,24]. For these reasons, choosing the most appropriate decompression method can be challenging given the need to balance short- and long-term outcomes. Moreover, guidelines on this topic are inconsistent about the optimal treatment to choose [10,25,26].

Several randomized trials investigated this issue and were summarized in several meta-analyses [21,27,28,29,30,31,32,33,34,35], which compared short- and long-term outcomes of ES and stenting as a bridge to surgery in malignant LBO (Table 1). Considering short-term outcomes, multiple studies demonstrated that post-operative morbidity, such as the rate of anastomotic leak and wound infection, was significantly lower in patients who underwent stenting as a bridge to surgery [27,29,31,32,33,35]; among these, only one study also showed a significantly lower post-operative mortality [33] in this group of patients. Other short-term surgical outcomes that have a significant impact on patients’ quality of life, such as the rate of temporary [27,29,32,33] or permanent [27,29,35] stoma, were significantly lower in the group of stenting as a bridge to surgery with an odds ratio (OR) of 0.39 [33]. Finally, two meta-analyses [32,33] compared the rate of laparoscopic versus open resection in the two groups, finding a significantly higher rate of laparoscopic resection in the endoscopic stenting group. On the other hand, the implantation of an endoscopic stent may increase the length of stay in order to wait for the time for recanalization and optimal timing for surgery [29,32]; however, this hospitalization time is usually exploited for the correct clinical staging, for patients’ stabilization and restarting of enteral nutrition.

While short-term outcomes are globally in favor of the bridge-to-surgery approach, less evidence supports this kind of approach for long-term outcomes. Different meta-analyses failed to find any differences between ES and stenting as a bridge to surgery in terms of overall survival [21,28,30,31,32,34,35] or disease-free survival [21,28,30,31,32,34] at three or five years. Moreover, the meta-analysis by Arezzo et al. found a higher local recurrence rate in the stent as a bridge to surgery group that was not even statistically significant (40.5% vs. 26.6%, *p* = 0.09) [29]. However, several other meta-analyses did not confirm this finding [21,28,30,31,32].

In conclusion, the current available evidence on the role of endoscopic stenting as a bridge to surgery are globally weak. However, short-term outcomes such as the rate of permanent or temporary stoma, which significantly affect patients’ quality of life, are significantly improved with the endoscopic stenting. It is therefore recommended to propose the endoscopic stenting as a first line of treatment in this situation only if the global organization of the Institution (i.e., on-call endoscopist with expertise in radiological procedure availability 24 h a day) and an agreed pathway with surgeons allow this kind of approach. As an alternative, ES remains a valid option for the treatment of resectable patients with acute malignant colonic obstruction.

## 3. Pre-Operative Evaluation: How the Radiologist Can Help the Endoscopist

### 3.1. Diagnosis of Large Bowel Obstruction

Clinical suspicion of bowel obstruction can be confirmed by various imaging methods that assist radiologists in addressing critical issues, such as identifying the specific location and underlying reason for the blockage, as well as determining the presence of any associated complications.

Plain abdominal radiography is usually first prescribed in the emergency department in patients with a suspicion of acute bowel obstruction because of its speed of acquisition, low cost, wide availability and low radiation exposure. Plain abdominal radiography, in the dependent and nondependent position, can provide diagnostic confirmation in approximately 50–70% of cases [36]. Typical radiographic indicators of LBO include enlargement of the colon and cecum, with diameters greater than 6 cm and 9 cm, respectively. Significant gas depletion in the rectum and accumulation of fecal material in the proximal colon is often observed [37]. In cases of LBO, the dilation of the small intestine may be variable, which depends on factors such as the duration of obstruction, the presence of a closed loop or a functional ileocecal valve [38]. In particular, when the ileocecal valve is incompetent in the context of a distal LBO, diffuse distention of the small and large intestines may mimic the appearance of pseudo-obstruction upon radiography. In this situation, distention of the large bowel is not marked and endoscopic stenting may not be organized as an emergent procedure. Moreover, temporary placement of a nasogastric tube could reduce bowel distension.

On the contrary, where a closed loop develops, the absence of proximal decompression increases the risk of progressive and localized colonic dilatation, ischemia and perforation. In this situation, the addition of computed tomography (CT) is necessary in order to obtain further information on the possible development of complications. A meta-analysis reported a CT sensitivity of 92% (range 81–100%) and specificity of 93% (range 68–100%) in detecting complete obstruction [39]. An abdominal CT scan is also necessary in order to investigate the etiology of acute obstruction.

### 3.2. Differential Diagnosis and Characteristics of Malignant Acute Colonic Obstruction from CT Scan

Multiplanar thin-layer acquisitions could delineate precisely the morphology of the colon, allow for the diagnosis of intraluminal, mural, and extramural causes of LBO and detect any potential complications. Performing a CT scan also in the urgent setting of malignant acute colonic obstruction also offers the advantage of detecting local and distant metastases. Intravenous contrast medium is recommended to help in identifying the presence of a mass and signs of inflammation and/or ischemia of the bowel wall. Intravenous iodinated contrast agent is essential to evaluate the enhancement of the colonic wall and can be administered with a weight-based protocol and a rate of 3 mL/s [40]. Indeed, an unenhanced CT scan for the evaluation of abdominal pain in the emergency department has an accuracy 30% lower than an enhanced CT scan [41] (Figure 1).

Coronal and multiplanar reformulations allow for the identification of the course of the distended bowel and the exact location of the obstruction, particularly in the case of rectosigmoid carcinoma where the distance of the stricture from the anal sphincter can be determined with good accuracy.

The diagnosis of LBO is primarily based on the presence of a dilated large bowel proximal to the colonic wall thickening with luminal narrowing and decompressed bowel distal to the stricture. The identification of a transition point is considered a clear sign for the diagnosis of colonic obstruction. CT-scan signs of malignant colonic obstruction are a localized thickening of the colonic wall with an uneven pattern, resembling an “applecore”, or the presence of a contrast-enhanced soft tissue mass situated at the center of the colon, causing a narrowing of the colonic passage [42]. In rare instances, a central necrosis or the presence of air within the lesion resembling an abscess-like lesion may also be observed.

It is important to consider the possibility of diverticulitis as a differential diagnosis, especially when there is evidence of colon cancer extending into the surrounding adipose tissue. Diverticulitis is typically represented by segmental, symmetric bowel wall thickening with hyperemia, the presence of fluid in the root of the mesentery and vascular engorgement [43]. Additional indications of active disease include inflammation of the mesenteric fat, the presence of abscesses and the formation of phlegmon. In a retrospective study conducted by Chintapalli et al., they examined all potential CT signs that could aid in distinguishing diverticulitis from CRC. They concluded that the most specific findings for diverticulitis were pericolonic stranding and a length of the involved segment of more than 10 cm, whereas the presence of pericolic lymphadenopathies and presence of a luminal mass were more commonly found in CRC [42].

To summarize, to enable endoscopists and surgeons to choose the best therapeutic strategy, the radiologist should make a diagnosis of a likely malignant colonic lesion, indicate the site and length of stenosis and its distance from the anal sphincter (in the case of rectal stenosis), and identify the presence of any complications (especially perforation and ischemia). Post-stent radiological control is not necessary except for cases where stent migration is suspected and plain radiography is sufficient.

## 4. Influence of Chemotherapy on Endoscopic Colonic Stenting: The Oncologist Point of View

Patients treated with stenting, especially in the palliative setting, may also experience adverse events related to chemotherapy or tumor progression. Delayed stent migration may occur under chemotherapy due to tumor regression, whereas re-obstruction or perforation could happen in relation to tumor growth [44]. However, the risk of long-term complications must be weighed against the lower mortality and earlier start of chemotherapy compared to surgery. Several small retrospective studies have shown that colonic stenting, compared to ES, reduces the duration of a hospital stay and allows the earlier initiation of chemotherapy, thus showing an advantage in quality of life and overall survival [14,17,45].

### 4.1. Timing and Safety of Chemotherapy Initiation in Patients with Acute Colonic Malignant Obstruction

Karoui et al. conducted a retrospective study with 58 stage IV obstructive CRCs comparing stent insertion or surgery as a palliative treatment. They demonstrated that the median time to chemotherapy beginning was shorter after stent insertion than after ES (14 vs. 28.5 days) with a potential benefit in quality of life and survival [17]. Lee et colleagues analyzed 88 patients with occlusive stage IV CRC who underwent surgery or stent insertion. In this study also, the median chemotherapy initiation was shorter in the stent group (8.1 days vs. 21.7 days) than in the surgery group [45]. In a meta-analysis, Zhao et al. considered 837 patients, of which 404 were treated with colonic stent and 433 with surgery. They concluded that the median time to start chemotherapy was significantly lower in the stent group than in the surgery group (15.5 days vs. 33.4 days) [14].

When determining the appropriateness of stent placement, the clinician should consider the risk of long-term stent-related complications in relation to the lower short-term mortality and faster onset of systemic chemotherapy treatment, considering that survival advantage with chemotherapy could expose patients with a colonic stent to an increased risk of long-term complications [14]. However, a multicenter retrospective study evaluating the outcomes of palliative chemotherapy without target therapy after colonic stent insertion for an obstructive primary tumor demonstrated that this is feasible and safe. The response rate (38%) and the disease control rate (62%) with Folfox or Folfiri were similar to other previous clinical trials using similar regimes, and the perforation rate after 2–15 months of placement was acceptable [46].

### 4.2. Effects of Antiangiogenic Agents following Colonic Stenting

Bevacizumab is a recombinant humanized monoclonal antibody that blocks the activity of the vascular endothelial growth factor, which is well-known to be a risk factor of intestinal perforation, which occurs in 1–2% of the patients [6]. Chemotherapy with anti-angiogenic agents showed a higher response rate and extended overall survival, thereby it is established as the first-line treatment in stage IV CRC. However, safety issues were raised in patients undergoing colonic stent placement, especially due to the risk of perforation. Several mechanisms have been proposed to explain the increased perforation rate in patients that received bevacizumab, such as tumor regression, necrosis, and a weakened serosa, combined with the pressure of the radial force of the stent on the colonic tumor [47].

Two retrospective analyses showed that bevacizumab increased the risk of perforation in patients with previous placement of a colonic stent. Small et al. collected data of stage IV CRC patients who had a stent placed for palliation or as a bridge to surgery. Eight patients of twenty-six (34.8%) reported a stent–related complication and a median of 44 days after bevacizumab started vs. a 22.8% complication rate in untreated patients. Four patients treated with antiangiogenic agents developed colonic perforation after 21 days of bevacizumab starting (15.4%) compared to a lower incidence in untreated patients (6.8%) [48]. The same result was obtained by Imbulgoda et al. who found a 20% rate of perforations in patients treated with chemotherapy and bevacizumab after stent placement [49]. A meta-analysis of 86 studies including 4086 patients with colonic stenting in malignant LBO showed that bevacizumab-based therapy was associated with a perforation rate of 12.5%. However, in this study only 2.1% of the included patients were treated with bevacizumab and the timing between the placement of the colonic stent and the start of antiangiogenic therapy was not specified [50].

Conversely, Lee and colleagues recently conducted a study of 104 patients who had received a colonic stent and were subsequently treated with bevacizumab. In the group treated with bevacizumab, only one patient had perforation as opposed to the three patients in the non-bevacizumab group [51]. Similarly, a relatively large retrospective study of 353 patients conducted by Park et al. demonstrated that the rate of perforation in patients with bevacizumab and without bevacizumab were equivalent (7.5% and 7%, respectively). Furthermore, the majority of the patients receiving bevacizumab who experienced perforation had started antiangiogenic treatment 1 month after stent insertion [19]. Fuccio et colleagues reported a retrospective case series of 91 patients who underwent palliative stent placement and they found that the complication rate was higher, though not statistically different, in the group treated with bevacizumab alone (29.4%) compared to those treated with chemotherapy alone (19.6%) or chemotherapy plus antiangiogenic agents (18%). It is also interesting to highlight that patients receiving chemotherapy and molecular-targeted drugs had longer overall survival compared to those receiving only chemotherapy, without showing an increased risk of stent-related complications [52]. Additionally, the study conducted by Pacheco-Barcia et al. on 78 patients demonstrated that chemotherapy plus antiangiogenetic agents improved overall survival compared to chemotherapy alone (43 months vs. 20 months) without an increase in risk perforation. The authors concluded that the increased risk of long-term stent complications in patients treated with chemotherapy and bevacizumab should not preclude the administration of these therapies for patients with metastatic colon cancer, considering the substantial response rate and its correlation with OS [53].

Nonetheless, due to the limited incidence, the lack of evidence, the retrospective nature of performed studies and their contradictory results, it is not possible to determine a definitive risk of stent-related perforation following bevacizumab treatment beginning [19,48,50,51]. Consequently, the most recent European guidelines suggest considering the use of bevacizumab following colonic stenting [10]. In clinical practice, colonic stenting serves as a life-saving procedure in acute LBO situations and it should be considered as a first option where local expertise is available, in elderly patients or in patients unfit for surgery or emergency surgery, regardless of the risk occurring with a potential therapy with bevacizumab [54]. Multidisciplinary discussion, weighing the potential benefits and harm for each patient, is therefore the decision-making moment in this acute situation.

No data are available about the risk of perforation for the new antiangiogenics recently prescribed in colorectal cancer (regorafenib, aflibercept). It could be speculated that considering the comparable mechanism of action with bevacizumab, the risk of gastrointestinal perforation induced by these new drugs is similar to that reported for bevacizumab.

### 4.3. Colonic Stenting during or after Antiangiogenic Treatment

Approximately 20% of patients with previous systemic chemotherapy develop primary tumor obstruction. In these patients, the benefits of minimally invasive SEMS over surgery would be greater since a high mortality rate and major complications following ES have been reported in this patient setting. Regarding the perforation risk of SEMS placement in patients with previous bevacizumab use, there are very limited evidence and clinical data.

Imbulgoda et al. collected the data of 87 stage IV CRC patients treated with or without chemotherapy and bevacizumab before or after stent placement. Perforation occurred globally in four patients (13%). Two out of the ten patients (20%) that received chemotherapy plus bevacizumab (3 before stenting, 6 after and 1 unknown) had a perforation. This result was not statistically different than the rate of perforation that occurred in patients treated with chemotherapy alone. However, the low number of events could have affected the statistical result. Patients who received chemotherapy with or without bevacizumab before stenting were treated for at least 50 days before the procedure and in this category of patients, the rate of perforation was the lowest reported (6%). It could be argued that a major determinant for perforation is the interval between stenting and chemotherapy infusion [49].

Bong et al. retrospectively collected the data of 1008 patients with metastatic colorectal cancer treated with bevacizumab and the incidence rate of complications requiring surgery was approximately 5.9% (60/1008). In this study, twenty-three patients during bevacizumab exposure required SEMS insertion and seven of these patients (7/23, 30.4%) experienced perforation requiring surgery. The authors concluded that SEMS was a significant risk factor for complication requiring surgery in patients already receiving bevacizumab [55]. In addition, in Park’s previously mentioned study, bevacizumab was administered both before and after colonic stenting. Only 2 out of 96 (2%) patients who developed perforation received stent implantation within 30 days before or after the start of bevacizumab treatment [19].

Based on these very low-quality data, ESGE guidelines suggest against colonic stenting while patients are receiving antiangiogenic therapy such as bevacizumab [10]. However, in this situation is fundamental to discuss with the oncologist whether the patient could benefit from a minimally invasive treatment (namely colonic stenting) and eventually try to extend the time between the last infusion of bevacizumab and colonic stenting.

## 5. Endoscopic Stenting

### 5.1. Preparation to Endoscopy

Colonic stenting, similar to other interventional endoscopic procedures, necessitates an adequate endoscopic setting in order to facilitate the procedure and minimize the adverse events (Figure 2). A room equipped with fluoroscopy is mandatory, whereas a radiology technician is not necessary if the endoscopic team is able to understand radiological images and to maneuver the C-arm.

A reasonable bowel preparation is suggested before the procedure in order to reach the stenosis quicker, to identify the residual lumen and to expedite the entire procedure. A standard bowel preparation is contraindicated in consideration of the occlusive status; thus, cleansing enemas are suggested before the procedure [56].

Considering the type of sedation to perform and the anesthesiologic support, it is important to evaluate patients’ comorbidities and the degree of respiratory distress related to the abdominal distension. In the case of compromised patients, especially those requiring non-invasive ventilation support, it is suggested to perform the endoscopic procedure in an equipped room with anesthesiologic support. If the patient is not in critical condition, colonic stenting is not a painful procedure and can be performed under conscious sedation with a combination of benzodiazepines and opioids.

Lastly, regarding which endoscopist could perform colonic stenting, it is necessary to consider that there is a learning curve for the endoscopist performing the stenting procedure. The level of expertise has a direct correlation to the rates of successful stenting and complications [57]. To increase the chances for success, the endoscopist is expected to have attempted more than 20 procedures and be familiar with other fluoroscopic endoscopic procedures like the endoscopic retrograde cholangiopancreatography (ERCP) [58].

### 5.2. Technique

The procedure requires an endoscope with a large operative channel, with a diameter of at least 3.7 mm, such as a standard colonoscope or an operative gastroscope. The presence of the carbon dioxide pump is also suggested in order to minimize abdominal distension and to prevent air dissemination in case of perforation [7]. Prophilactic antibiotics are not routinely suggested because the risk of bacteremia after stent insertion is very low [10].

The patient can be indistinctly supine or on the left side: the former facilitates the interpretation of radiological frames while the latter, which is the standard colonoscopy position, allows for an easier passage through the sigmoid colon. We usually start the procedure on the left side and once the stricture is reached, we evaluate radiologically the extent and location of the stricture with the contrastography obtained with air insufflation (Figure 3). If the radiological orientation of the stricture is not clear, we rotate the patient in the supine position in order to obtain a clearer radiological view of the stricture.

A long 0.035 inch guidewire (450 cm) with a hydrophilic tip is inserted through a cannula in order to pass the stricture. In case of angulated strictures, the use of a sphincterotome or an angle tip guidewire could help the passage of the stricture. Changing the patient’s position could sometimes help to better visualize the stricture. After the evaluation of the correct position and evaluation of the length of the stricture with the use of a contrast agent, the guidewire is pulled over to create safe loops over the tumor and the cannula is then retracted.

Histological confirmation of the malignant etiology of the stricture is mandatory [10]; however, the urgent setting and the location of the stricture could increase the difficulty of obtaining biopsies. Moreover, the post-biopsy bleeding could obscure the stricture visualization and consequently increases the difficulty of guidewire passage. What we usually suggest is to first put the guidewire over the stricture and then take the biopsies before stent placement in order to guarantee access at the stricture.

Colonic stent placement could be performed with the through-the-scope (TTS) technique or the over-the-wire (OTW) technique. Conflicting results have been produced about technical and clinical benefits between the two procedures [59]. However, with the widespread availability of TTS stents it has become a common practice to proceed with the TTS technique, which allows an endoscopic control during stent release, and to reserve the OTW technique when larger stents are needed or when an endoscope with a smaller operative channel is used instead of standard scopes for anatomical reasons.

The SEMS is then advanced over the guidewire under radiologic and endoscopic visualization. The cranial flange is the first opened and then retracted close to the cranial extreme of the stricture when a little feeling of resistance at the catheter is appreciated. Then, the entire stent is released. In the case of angulated strictures, it is suggested to maintain an adequate distance with the endoscope from the stricture and to pay attention during the release maneuver. In this situation, it is suggested to pull the SEMS catheter vigorously when the stent is released in order to avoid misdeployment of the stent in the proximal colon. Dilation of the stricture before or after the stent placement is contraindicated because a retrospective analysis evidenced a higher rate of colonic perforation [60].

Different stents are available for colonic stenting and all of them are composed by nitinol, a nonferromagnetic blend. Considering the type of the stent chosen, some meta-analyses evidenced the superiority of uncovered stents compared to covered ones in terms of global complications, tumor overgrowth and stent migration resulting in longer stent patency. Only the risk of tumor ingrowth is reduced with covered SEMS. Partially covered stents potentially could reduce the risk of migration; however, no studies found a superiority of this type of stents compared to uncovered or fully covered stents [61]. The most recent guideline suggests with a weak grade of recommendation to use an uncovered stent as a first line approach [62], however the stent choice should also be made on a clinical basis.

The length of the stent should be tailored to the length of the stricture and its position. It is generally suggested to extend the length of the stent at least 2 cm before and after the stenosis, in particular for tumors at flexures, for which the risk of migration is higher [10]. No definitive conclusions have been reached about the ideal stent diameter, but few studies reported a higher rate of stent migration with a diameter less than 24 mm [63,64].

Tumor localization requires careful consideration. Some authors have expressed reservations about the use of colonic stents for proximal colonic obstruction or rectal malignant stricture near to the anal verge. Studies comparing the clinical success rates between right-sided and left-sided stenting showed conflicting results. However, several retrospective studies evidenced the feasibility of the stenting of the proximal colon (Figure 4) [65,66,67,68].

In summary, in case of right-sided LBO it is important to evaluate the patency of the ileocecal valve. In the case of involvement of the ileocecal valve by the tumor, it is difficult to proceed with the stenting and have a clinical benefit. On the other hand, in the case of a patency of the ileocecal valve and identification of the stricture distally to the ileocecal valve, it is possible to attempt the placement of an SEMS after a multidisciplinary consultation.

Tumor location in the distal rectum, defined as a mass within 8–10 cm of the anal verge, raises concerns due to the potential post-procedural pain and increased risk of migration, tenesmus and bleeding. Data on rectal stenting are scarce, with only one systematic review and meta-analysis evaluating short-term outcomes. This study found that rectal and recto-sigmoid stenting had a high technical success rate, but this did not always translate into clinical success (97% and 69%, respectively). Additionally, the overall complication rate was 32%, with stent re-obstruction and migration being the most common issues (10.5% and 9.3%, respectively) [69]. Consequently, no definitive conclusions can be drawn from these data and in the case of acute colonic obstruction related to a tumor close to the anal verge, it is fundamental to establish the clinical indication case by case.

### 5.3. Procedural Adverse Events and Post Endoscopic Management

Complications arising from the insertion of SEMS can be divided into minor and major complications according to the degree of severity [70]. Perforation is the most common (4–7%) and most feared major complication that can occur during endoscopic stent placement [71,72]. Immediate causes of perforation include wire or catheter misplacement beyond the colonic wall. Delayed perforation usually occurs in the distended right colon when there is a thin walled caecum; this type of perforation usually occurs concomitantly to the placement of SEMS in a difficult endoscopic position that leads to an excessive amount of air insufflation in a distended large bowel [73]. Conservative treatment of perforation is possible when the SEMS is correctly placed and there is only a modest quantity of air in the abdomen without the presence of fluid and without clinical worsening. Otherwise, surgical treatment is the only therapeutic choice for colonic perforation. Lastly, late perforation can be caused by concomitant therapy with antiangiogenic agents, as mentioned before.

Stent migration is the other major complication that may occur during endoscopic positioning. In the case of distal migration, the stent could be retrieved endoscopically with caution in order to avoid mucosal damage and a new stent could be repositioned. Stent migration proximally to the stricture makes impossible the endoscopic retrieval of the stent. In the case of palliative indications, the migrated stent could be left in the proximal colon but clinical monitoring in search of signs of decubitus of the stent is needed [5]. Bleeding is a minor complication that may occur after stent insertion but it is usually self-limiting.

The clinical success of endoscopic stenting is obtained when there is an adequate passage of air and feces from the anus after the endoscopic procedure. Studies do not specify which is the ideal timing to evaluate an effective colonic recanalization. Usually, in the case of a correct stent deployment, the passage of air and of liquid feces is almost immediate and patients’ relief is evident. In the case of angulated (>165°) and long strictures, colonic canalization could require more hours. These types of strictures are also those that, despite a complete technical success (usually obtained in 95% of patients), have a short term clinical failure and therefore require surgery resolution of the obstruction [73].

The optimal interval between SEMS placement and subsequent surgery in patients with MBO is still unclear and few studies evaluated this question. The most recent guidelines, based on low-quality studies, and randomized trial suggest surgery after 5 to 14 days to the colonic stenting. In our Institution, in cooperation with surgeons we usually candidate the patient to surgery after a bridge with endoscopic stenting when there are no more signs of obstruction and it is possible to perform a complete bowel preparation [74,75].

Following colonic stenting for malignant LBO, it is necessary to perform a complete colonoscopy to exclude the presence of a synchronous lesion, which occurs in 3–13% of patients with CRC [76,77]. Through-the-stent colonoscopy is feasible after an adequate PEG-based bowel preparation, with a success rate ranging from 63% to 100%; notably, the lowest rate increased to 87.5% when a gastroscope was utilized instead of a colonoscope [77,78]. It is important to state that through-the-stent colonoscopy is a safe procedure with no instances of migration or bleeding related to this implementation [77,78]. There is a complete lack of research concerning the optimal timing for proximal colonic evaluation after the revealing of a CRC. For these reasons, the most recent ESGE guidelines suggest conducting a complete colonoscopy no more than 6 months after colonic stenting and surgery or even before the surgery in the curative setting [17].

## 6. Conclusions

Acute malignant colonic obstruction is an urgent clinical condition that still has some grey areas regarding endoscopic management. Robust data support colonic stenting in the palliative setting [11,12,13,14,15,16,17,18] and demonstrate improved quality of life and better clinical outcomes compared to ES. However, such robust scientific support is lacking for other more challenging situations such as resectable patients who may benefit from the bridge-to-surgery approach and those experiencing acute obstruction under antiangiogenic treatment. The available scientific data show that colonic stenting certainly improves the short-term outcomes of patients who undergo the bridge-to-surgery approach and that endoscopic stenting close to bevacizumab administration may increase the risk of perforation [19,27,29,31,32,33,35,49,55]. The location of malignant stricture in debated positions, such as in the right colon or close to the anal verge, may limit the clinical success of colonic stenting. The literature suggests that there are no absolute contraindications to colonic stenting even in these technical and clinically challenging situations and a minimally invasive treatment compared to ES could improve patients’ quality of life.

Acute malignant colonic obstruction is therefore a critical situation with some challenging clinical and technical issues that frequently necessitates real-time consultation among different specialists to establish the most suitable therapeutic strategy. In our opinion, we suggest organizing a local pathway shared with the radiologist, surgeon, oncologist and endoscopist to create a predefined algorithm based on the available local resources. For instance, as highlighted in the present review, it is fundamental to perform an enhanced CT scan in the diagnostic process of LBO to establish its etiology and to study the extension of the stricture in preparation for endoscopic stenting. However, many non-tertiary centers do not have the 24 h availability of enhanced radiology and this aspect could restrict the endoscopic stenting. The absence of specialized figures that could approach patients with acute colonic obstruction could limit endoscopic treatment in favor of surgery. Moreover, the local diagnostic-treatment algorithm should consider the number of available on-call endoscopists to perform colonic stenting. Lastly, it is reasonable that in more structured institutions with a high availability of human resources and expertise, the clinical indications for endoscopic stenting may be expanded to include borderline situations (e.g., bridge-to-surgery indication, challenging endoscopic positions).

## Figures and Tables

**Figure 1 cancers-16-00821-f001:**
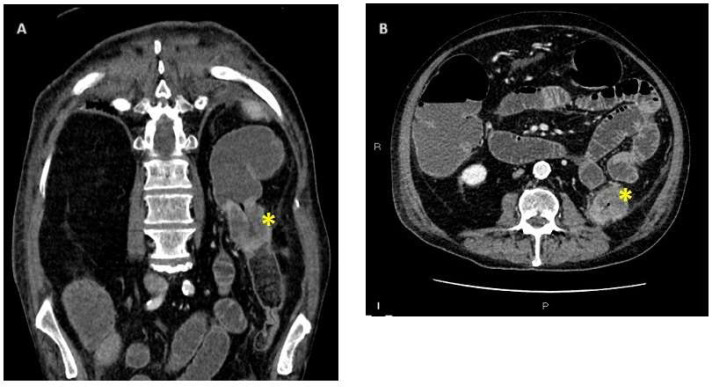
Enhanced CT scan of acute colonic obstruction with presence of neoplastic stricture at the level of the descending colon. In the coronal section: (**A**) stenosis of the descending with marked thickening of its walls, non-homogeneous contrast enhancement, possible extra-visceral extension; in the ax, (**B**) overdistention of large and small bowel with air fluids levels. Yellow stars at the level of the neoplastic stricture.

**Figure 2 cancers-16-00821-f002:**
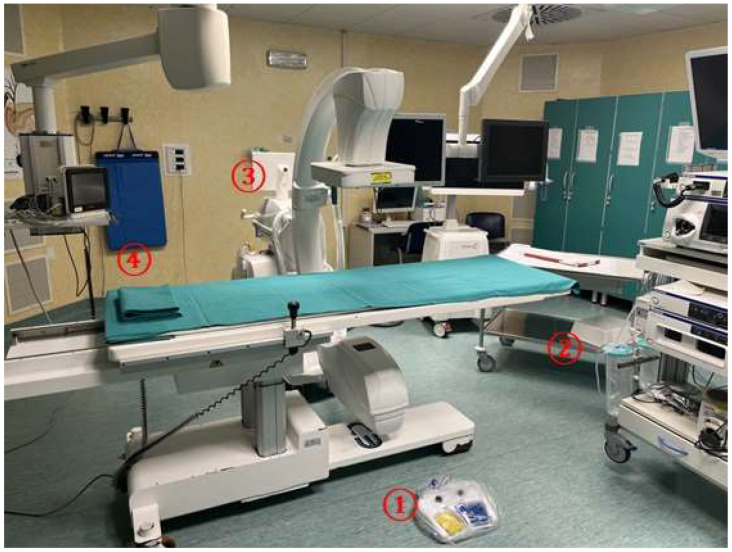
Example of an endoscopic setting for the placement of a colonic stent. ① Endoscopist position with radiological and endoscopic monitor placed frontally and the endoscopic processor in the right position. ② Nurse position at the end of the radiological bed with an instrument shelf on his side. ③ Position of the assistant at the C-arm. ④ Anesthesiological position at the patient’s head with vital monitoring on his side.

**Figure 3 cancers-16-00821-f003:**
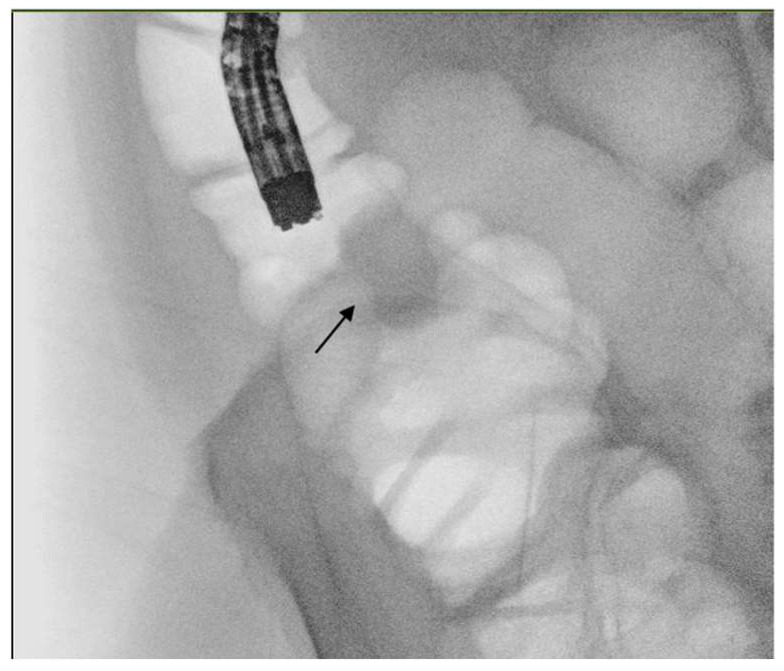
Contrastography obtained with air insufflation showing colonic stricture (black arrow) at the level of ascending colon.

**Figure 4 cancers-16-00821-f004:**
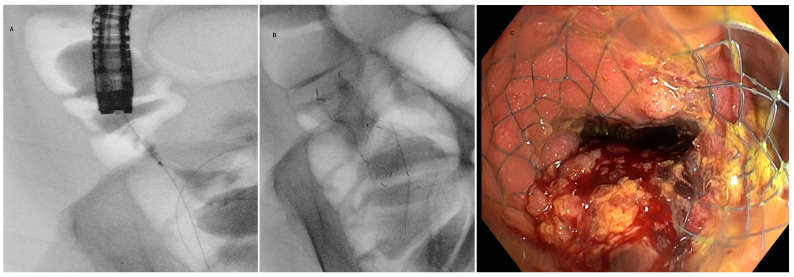
Colonic stenting of a malignant stricture at the level of the ascending colon. (**A**): Contrast injection at the level of the stricture with loop of the guidewire in the cecum; (**B**) radiological evidence of the self-expandable metal stent (SEMS) within the malignant stricture; (**C**) endoscopic view of the stricture with the metallic mesh of the distal part of the uncovered SEMS.

**Table 1 cancers-16-00821-t001:** Short- and long-term outcomes of Emergency Surgery and Stent as Bridge to Surgery (SBTS) in malignant bowel obstruction reported in the most recent meta-analyses. Green color: outcomes in favor of SBTS; red color: outcomes in favor of ES; yellow color: outcomes in which ES and SBTS are not significantly different.

Meta-Analyses		Short-Term Outcomes						Long-Term Outcomes		
	Included Studies	Morbidity	Mortality	Stoma Rate	Permanent Stoma	VLS	LOS	OS	DFS	Local Recurrence
Allievi 2017, [27]	7 RCTs (448 pts)	ES 54.8% vs. SBTS 37.8% (*p* = 0.02)	ns	ES 46.0% vs. SBTS 28.9% (*p* < 0.0001)	ES 35.2% vs. SBTS 24.5% (*p* < 0.02)	/	/	/	/	/
Amelung 2018, [28]	5 RCTs, 4 prospective, 12 retrospective (1919 pts)	/	/	/	/	/	/	3 years and 5 years: ns	3 years and 5 years: ns	ns
Arezzo 2017, [29]	8 RCTs (497 pts)	ES 51.2% vs. SBTS 33.9% (*p* = 0.023)	ns	ES 51.4% vs. SBTS 33.9% (*p* < 0.001)	ES 35.2% vs. SBTS 22.2% (*p* = 0.003)	/	ES 14.5 d vs. SBTS 15.5 (*p* = 0.039)	/	/	ES 26.6% vs. SBTS 40.5% (*p* = 0.09)
Cao 2019, [30]	5 RCTs, 3 prospective, 16 retrospective (2508 pts)	/	/	/	/	/	/	3 years and 5 years: ns	3 years and 5 years: ns	ns
Ceresoli 2017, [21]	5 RCTs, 3 prospective, 9 retrospective	/	/	/	/	/	/	ns	Ns	ns
Foo 2019, [31]	7 RCTs (448 pts)	Reduced in SBTS (RR 0.6, *p* = 0.032)	ns	/	/	/	/	3 years: ns	3 years: ns	ns
Jain 2020, [32]	8 RCTs, 25 observational (15,224 pts)	SBTS vs. ES: anastomotic leak OR 0.59 (*p* = 0.006), wound infection OR 0.64 (*p* = 0.004)	SBTS vs. ES: OR 0.69 (*p* = 0.010)	SBTS vs. ES: OR 0.39 (*p* < 0.001)	ns	SBTS vs. ES: OR 5.9 (*p* < 0.001)	SBTS vs. ES: Reduction in ES (<0.001)	ns	Ns	ns
Kanaka 2022, [33]	Right-sided MLBO:#break#7 observational studies (5136 pts)	SBTS vs. ES: OR 0.78 (*p* = 0.003)	SBTS vs. ES: OR 0.51 (*p* = 0.03)	SBTS 2.0% vs. ES 11.0% (*p* < 0.01))	/	SBTS 48.5% vs. ES 15.7% (*p* < 0.01)	/	/	/	/
Matsuda 2015, [34]	2 RCTs, 9 observational studies (1136 pts)	/	/	/	/	/	/	3 y and 5 y: ns	3 y and 5 y: ns	/
McKechnie 2023, [35]	Network meta-analysis: 53 studies	ES vs. SBTS: OR 2.14 (*p* < 0.0019	ns	/	ES vs. SBTS: OR 2.91 (*p* < 0.001)	/	/	3 y and 5 y: ns	/	/

ES, emergency surgery; SBTS, stent as a bridge to surgery; VLS, videolaparoscopy; LOS, length of stay; OS, overall survival; DFS, disease-free survival; RCT, randomized controlled trials; pts, patients; OR, odd ratio.

## Data Availability

No new data were created or analyzed in this study. Data sharing is not applicable to this article.

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
