# Peer review of "Malignant Acute Colonic Obstruction: Multidisciplinary Approach for Endoscopic Management"

_cancers, 2024, doi:10.3390/cancers16040821_

Round 1

Reviewer 1 Report

Comments and Suggestions for Authors

Nice review, well written manuscript.

1) Cleaning enemas before all procedures, what is the risk of enema in symptomatic large bowel obstruction?

2) What is the role of prophylactic antibiotic ?

3) Colonic stent insertion can it be performed via either endoscopic or fluoroscopic method?

4) Which is advisable, only air contrastography, injection of water soluble contrast injection or both contrast and air injections?

5) Is it advantageous to put a partial covered stent?

Author Response

Thanks for your nice and constructive comments. Please find below our point-by point response:

  • To the best of our knowledge there are no data indicating that cleansing enema are contraindicated in patients with acute colonic obstruction. The only adverse event reported in literature was the rectal perforation related to iatrogenic mechanical damage. However, this type of adverse event is usually related to the application of the enema by non-professional figures and not by the clinical condition of acute obstruction.
  • Prophylactic antibiotics are not indicated by the most recent ESGE guidelines. We added a brief comment in the text (line 417-418)
  • We described both methods in the section 5.2 Techniques ( line 443-449) when we described the TTS and the OTW techniques
  • We usually use both methods to analyze the length of the stricture as described in line 419-415
  • Thanks for your interesting comment. To date, only uncovered stents showed a superiority over the other stents in terms of global reduced adverse events. The partially covered stent potentially reduces the risk of migration however the risk of tumor overgrowth is the same of covered stent. We added a comment in the text (line 463-68)

Reviewer 2 Report

Comments and Suggestions for Authors

It’s better to describe more about SEMS for colonic obstruction/stenosis due to metastatic disease (e.g., peritoneal metastasis, direct invasion from pancreatic cancer) in the manuscript.  

Regarding endoscopic procedure, 

(1)   Is not antibiotics mandatory before SEMS placement? 

(2)   We often use marking clip to visualize the tumor anal margin on fluoroscopy during SEMS placement. Is this not standard anymore?

(3)   Passing guidewire through a bulky tumor is the most challenging part of SEMS placement. Would you give us any tips to facilitate passing guidewire during SEMS placement in the manuscript?   

Comments on the Quality of English Language

No major problem

Author Response

Thanks for your constructive comments. Please find below our point-by-point response.

  • Our review was mainly focused on colonic obstruction because usually extrinsic strictures cause milder symptoms than acute primary colonic obstruction. Moreover, the clinical context of extrinsic compressions is more complex and usually requires a case-by-case discussion. We added a paragraph on this topic in lines 98-104
  • Prophylactic antibiotics are not indicated by the most recent ESGE guidelines. We added a brief comment in the text (line 417-418)
  • In case of TTS stents it is no more useful to put a radio-opaque marker (clips or external metallic small objects) because stent release is performed under endoscopic vision. Also in case of OTW technique the stricture is visualized with the previous administration of contrast within the stricture. However, there are no evidences for these kinds of intraprocedural tricks and if the operator is confident with the placement of metallic clip I think that he could continue with this kind of method
  • In line 415-416 we suggested the use of a sphincterotome instead of a cannula or a angle tip guidewire to help the passage of the guidewire through the stricture. We added also the changing patients position that could help the visualization of the stricture

Reviewer 3 Report

Comments and Suggestions for Authors

This is an interesting paper considering the current tendency for minimal invasive treatments.

However, the level of evidence for the treatment of malignant acute colorectal obstruction by endoscopic stenting as an alternative to surgical treatment is scarce. This should be clear and emphasized in the manuscript in order to avoid misinterpretation.

The authors highlight the need for accurate radiologic imaging for the diagnosis and procedure planning. The need for histological confirmation of malignancy is not mentioned at all. This should be commented and included in the manuscript.

The need for an expert team of medical specialties necessary to employ this policy and its availability around the clock should be emphasized. Apparently this is a huge restriction around the world for the adoption of endoscopic stenting as an alternative to surgery in real life.

Finally, there are several points that need linguistic (lines 22, 133, 165, 193, 205, 286, 287, 301, 303, 318, 319, 340, 342, 345, 355, 519, 533) and terminology attention and correction (lines 203 coronal sections, 391, 405-406, 409, 452, 459, 481, 486, 487, 505)

Comments on the Quality of English Language

Language editing is necessary, see linguistic (lines 22, 133, 165, 193, 205, 286, 287, 301, 303, 318, 319, 340, 342, 345, 355, 519, 533) and terminology  (lines 203 coronal sections, 391, 405-406, 409, 452, 459, 481, 486, 487, 505) points

Author Response

Thanks for your constructive comments. Please find below our point-by-point response:

  • We think that strong evidences in favor of endoscopic stenting are available in literature in the palliative setting. We agree with your comment for the bridge to surgery setting where the evidences are weak. We added a comment in the text (lines 162-170)
  • Thanks for your comment. Histological confirmation is necessary for all obstructive colonic tumors. However, the urgent setting could increase the difficulty to take the biopsies. We added a comment in the text. (line 421-26)

  • We strongly agree with your comment. In the conclusions we emphasized the concept that this kind of problem need an agreed pathway with the different specialistic figures and also the need of adequate expertise.

  • Thanks for your precise linguistic revision. We modified the linguistic errors.